# *Aedes aegypti* Hemocytes Mediate Antiviral Immunity

**DOI:** 10.3390/ijms26188779

**Published:** 2025-09-09

**Authors:** Victor Cardoso-Jaime, Chinmay V. Tikhe, Mihra Tavadia, George Dimopoulos

**Affiliations:** W. Harry Feinstone Department of Molecular Microbiology and Immunology, Johns Hopkins Bloomberg School of Public Health, Baltimore, MD 21205, USA; v.cardoso.jaime@gmail.com (V.C.-J.); ctikhe1@jhu.edu (C.V.T.); mihra.tavadia@ird.fr (M.T.)

**Keywords:** hemocytes, mosquito, dengue, RNAi pathways, virus, dissemination, basal lamina, phagocytosis, systemic infection

## Abstract

Hemocytes play several key roles in the mosquito’s immune response. Despite most of our understanding regarding their immunological role concerns their responses against bacteria, fungi, and *Plasmodium*, our knowledge of hemocyte’s role in antiviral defense is poorly understood. We performed a comprehensive comparative transcriptomic analysis between the dengue vector *Aedes aegypti*’s two major immune cell types, hemocytes and fat body, revealing a plethora of differentially expressed immune genes that indicates a high level of functional specialization as well as complementation between the two immune cell types. Our transcriptomic approach yielded molecular insights into the antiviral immune response of *Ae. aegypti* hemocytes during systemic infection. In fact, hemocytes showed abundant expression of RNAi pathway genes under naive conditions and upregulated many of these upon dengue virus (DENV) infection. Furthermore, chemical depletion of phagocytic hemocytes resulted in a higher DENV systemic infection. Our results suggest that hemocytes possess mechanisms to control systemic viral infections.

## 1. Introduction

Mosquitoes can be considered the world’s deadliest animal because they transmit a number of diseases, including malaria and arboviral diseases. The risk of a pandemic resulting from an emerging or re-emerging arbovirus, such as dengue, Zika, yellow fever, and chikungunya viruses, is expected to continue increasing due to human movement, urbanization, climate change, and mosquitoes’ adaptation have resulted in an uncontrolled expansion of the vectors [1]. Dengue has become the most widely expanded vector-borne viral disease worldwide, with the number of cases increasing over 30-fold in the last 50 years [2]. Over 300 million cases per year are estimated, and the factors outlined above will substantially increase the percentage of the population at risk of contracting dengue fever over the next 50 years [3,4].

While about 70 different species of mosquitoes are known to transmit malaria [5], dengue fever is mainly transmitted by *Ae. aegypti* [6]. Despite the high prevalence of dengue, only 0.05–10% of mosquitoes in a dengue-endemic area are found to be infected with arboviruses [7]. This low percentage is largely due to mosquitoes’ robust immune systems, which provide protection against bacteria, fungi, protozoa, and viruses. Mosquito immune systems, however, lack adaptative immune responses mediated by lymphocytes and antibodies. Instead, mosquitoes rely solely on their innate immune systems, which mediate melanization, the production of a variety of effector proteins such as antimicrobial peptides and other mechanisms. The mosquito’s antiviral immune defense is predominantly mediated by the RNA interference (RNAi) pathways, which depend on nucleases and RNA to target and degrade the viral genome. There are three RNAi pathways in mosquitoes: small interfering RNA (siRNA), micro RNA (miRNA), and P-element-induced wimpy testis (PIWI)-interacting RNA (piRNA) [8].

The innate immune responses also include cellular immune responses, such as phagocytosis, nodulation, and encapsulation [9,10], which are mediated by hemocytes. Mosquito hemocytes are classified by their morphology and functions into three subpopulations, prohemocytes, oenocytoids, and granulocytes [11,12]. In addition to mediating cellular immune responses, hemocytes are considered the exclusive producers of prophenoloxidases (PPOs) which are responsible for pathogen melanization, and they are the primary producers of many anti-*Plasmodium* factors, such as thioester-containing protein 1 (TEP1), *Anopheles Plasmodium*-responsive Leucine-rich repeat protein 1 (APL1C), Fibrinogen-like immunolectin 9 (FBN9), and others [13,14,15,16,17]. However, although many of the mechanisms of hemocyte immune responses against bacteria, fungi, and protozoa have been described, their role in the antiviral response has received less attention.

Interestingly, several studies have shown that many immune factors involved in the antiviral response in mosquitoes are mainly produced by hemocytes (detailed in [18]), suggesting they may play an important role controlling viral infections. Here, we have compared the transcriptomic profiles of fat body and hemocytes from non-infected *Ae. aegypti* mosquitoes and found that hemocytes have a higher basal expression of most known antiviral immune genes. In contrast, fat bodies express most of the CLIP-domain proteases, serpins, lectins, and leucine-rich repeat (LRR) proteins. Furthermore, we also show that phagocytic hemocytes are primarily responsible for controlling systemic infection with dengue virus, most likely through the RNAi pathway.

## 2. Results

### 2.1. Hemocytes Highly Express piRNA Pathway Genes

Several studies [19,20,21,22] have previously shown that hemocytes are implicated in antiviral defenses; however, the mechanisms involved in controlling viral infections are largely unknown. To address this knowledge gap, we performed a transcriptomic analysis of DENV-infected and un-infected *Ae. aegypti* hemocytes and fat body, the major immune tissues and cells in insects.

First, we performed a comparison between the naïve hemocyte and fat body transcriptomes (Figure 1a). This analysis revealed 3493 genes with higher mRNA abundance in hemocytes and 2764 with higher mRNA abundance in the fat body (Figure 1b and Appendix A). The gene set enrichment analysis showed that hemocytes were highly expressing genes involved in the dorso-ventral axis formation pathway, a category that also includes genes of the major piRNA pathway (Figure 1c and Appendix A). A detailed analysis of immune genes showed that hemocytes highly expressed genes associated with the Janus kinase/signal transducer and activator of transcription (JAK/STAT), c-Jun N-terminal kinase (JNK), mitogen activated protein kinase/extracellular signal-regulated kinase (MAPK/ERK), Transforming growth factor β (TGF-β), and nuclear factor κβ (NF-κβ) pathways, as well as genes involved in tissue structure and differentiation, such as laminins, collagens, and Delta/NOTCH. As expected, PPO genes were also highly expressed in hemocytes. Interestingly, some of the most representative categories of highly expressed genes in hemocytes were components of the autophagy, and RNAi pathways (piRNA, miRNA, and siRNA pathways) (Figure 1d). In contrast, the fat body showed high expression of other immunity gene families including CLIP-domain proteases, LRR proteins, lectins, and serine protease inhibitors (serpins) (Figure 1d). These results indicate that the major immune tissues, the fat body and hemocytes, play specialized immune functions.

A gene family enrichment analysis also identified genes that were highly expressed in hemocytes, which included PPOs, Serpin 10, and TEP1 [11,17,23]. In addition, we identified eight piRNA pathway genes, including Zucchini, mael, Atari, REXO5, and the antiviral genes PIWI2-4 and Argonaute 3 (AGO3) [24]; three genes of the miRNA pathway (loqs, Ago1b, and Ars2); one of the siRNA pathway (loqs2); and TOLL6, which has been suggested to be the receptor for the recognition of dsRNA produced during viral replication [25] (Figure 1e). Next, we performed an in silico analysis of interactions between these genes using the Search Tool for the Retrieval of Interacting Genes/Proteins (STRING). Interactions were predicted for most genes; however, the piRNA pathway genes PIWI4 and AGO3 showed the most interactions with other genes of the pathway (Figure 1f), suggesting that they play a central role in the piRNA pathway. In addition, we identified another group of interacting genes that includes Ago1b, loqs, and loqs2, which are essential components of the miRNA and siRNA pathways. Interestingly, all these genes play a key role in the antiviral immune response, suggesting that hemocytes may use these pathways to control viral infections.

### 2.2. DENV Infection Upregulates Genes of the RNAi Pathway in Hemocytes

To identify the genes involved in hemocyte antiviral immune responses, we compared the transcriptomes of hemocytes from DENV-infected and naïve mosquitoes. (Figure 2a). We constructed transcriptome libraries from hemocytes collected 10 days post-infection, a time point when most studies indicate that the virus has infected most secondary tissues, including hemocytes [26,27,28]. This timing ensures that hemocytes have come into contact with the virus and have likely initiated an immune response. The analysis showed 277 upregulated and 65 downregulated genes in infected hemocytes (Figure 2b and Appendix A). Some of the most enriched sets of genes (based on KEEG pathway enrichment) were related to ribosome spliceosomes, the mechanistic target of rapamycin (mTOR) signaling pathway, the Wingless (Wnt) signaling pathway, and the mRNA surveillance pathway (Figure 2c and Appendix A). We found that genes related to ribosomes, retrotransposons, cytochromes, C-type lectins, the cell cycle, DNA repair, and miRNA and piRNA were strongly upregulated in DENV-infected hemocytes **(**Figure 2d). However, some genes related to cell cycle, DNA repair, and spliceosome genes were downregulated (Figure 2d). Interestingly, the *Drosophila* miRNA pathway genes scaffold attachment factor B2 (SAF-B2), the Cap-binding protein 20 (Cbp20), and the mosquito PIWI5 gene that play an antiviral function in the piRNA pathway were also highly enriched [29,30,31]. Moreover, retrotransposons have also been shown to play roles in the antiviral immune responses of mosquitoes [32,33]. Altogether, these results suggest that hemocytes express a plethora of antiviral immune genes and may therefore play an important role in controlling viral infections.

### 2.3. Phagocytic Hemocytes Control Systemic Infection with DENV

Based on our RNAseq data which shows that hemocytes highly express antiviral immune genes, we hypothesized that hemocytes may be involved in controlling DENV infection. To test this, we depleted phagocytic hemocytes using a macrophage depletion kit, as previously described [34] to have achieved a 90% depletion (Appendix A). First, we evaluated whether depleting phagocytes could influence DENV infection of the midgut. We fed the non-treated mosquitoes (NT), control liposomes (LPSMs)- and liposomes-containing clodronate (CLDS)-injected mosquitoes on DENV-containing blood and evaluate the viral load in the midgut at 7 days post-infection. No effects of phagocytic hemocyte depletion were observed in terms of the intensity or prevalence of infection of the midguts (Figure 3a,b). An infected mosquito becomes infectious when the virus crosses several physical and immunological barriers and is disseminated from the midgut to other tissues [35,36]. To minimize these extrinsic effects and thereby more accurately evaluate the role of phagocytic hemocytes in controlling systemic infections, we injected the same amount of DENV PFUs into the hemocoel and evaluated the viral load in the whole bodies of NT-, LPSMs-, and CLDs-injected mosquitoes at 7 days post-injection. As expected, phagocyte-depleted mosquitoes showed higher viral titers than did controls (Figure 3c,d), suggesting that hemocytes control systemic virus infection, but not viral infection of the midgut.

In *Drosophila*, hemocytes control midgut epithelium integrity and also control intestinal infections [37]. Interestingly, in mosquitoes, blood feeding disrupts the midgut epithelium, and viruses use these breaches as a means of escape into the hemolymph, from which they can spread to other tissues [38]. In *An. gambiae*, phagocyte depletion results in a high mortality rate after blood feeding because of the disruption of the midgut epithelium [39]. We therefore performed Smurf assays by feeding mosquitoes on blood containing blue dye in order to determine whether phagocytic hemocytes, in some way, could influence the midgut epithelium’s integrity. When the epithelium is disrupted, the blue dye is disseminated through the body, and the mosquitoes become blue (the “Smurf phenotype”) [40]. However, in agreement with our previous result indicating that hemocytes do not influence viral infection of the midgut, when infection was performed through blood feeding, no differences in mortality were observed between phagocyte-depleted mosquitoes and the controls, and Smurf mosquitoes were rarely observed (Appendix A), suggesting that phagocytes do not control midgut epithelium integrity in *Ae. aegypti*.

### 2.4. Hemocytes and Tracheal Structures Internalize Virus-Sized Nanobeads

Even though antiviral immune responses have been well studied in mosquitoes, the mechanisms of virus dissemination have received less attention. Mosquito tissues are covered by a basal lamina, which is permeable to particles smaller than 10 nm [41,42]. The DENV particle size is about 50 nm, suggesting that the basal lamina acts as a barrier, potentially limiting viral dissemination to tissue cells [38,43,44]. Interestingly, hemocytes and probably some tracheal structures are the only mosquito cell types lacking a basal lamina, which could facilitate the engulfment of viruses circulating in the hemolymph [18]. In addition, hemocytes are the sentinel cells capturing foreign particles circulating in the hemolymph such as bacteria, fungi, parasites [45,46], and most likely also viruses.

To test this hypothesis, we injected mosquitoes with 30 nm-diameter carboxylate-modified polystyrene latex fluorescent nanobeads, which have a similar size to the virus, to mimic the physical aspect of viral entry and assess tissue permissiveness. These particles have been extensively used to test tissue porosity and mechanisms of cell internalization in several cell types and animal models [47,48,49,50,51,52]. We tracked fluorescent nanobeads presence in hemocytes and their tissue dissemination at successive time points. We found that circulating hemocytes started to engulf the nanobeads as early as 2 h after injection. They reached the highest rate of engulfment at 4 h after injection, and by 24 h the nanobeads had almost disappeared from the hemolymph (Figure 4). The same pattern was observed in sessile hemocytes from the midgut, abdomen, and ovaries (Appendix A).

As expected, in addition to the hemocytes, we also observed nanobeads in the tracheolar structures on different tissues (Figure 5). This observation indicates that some tracheolar structures allow the particles to cross the basal lamina and most likely reach the epithelial cells. Like hemocytes, the presence of nanobeads could be tracked 2 h post-injection, and the maximum nanobead uptake occurred at 4 h (Appendix A), suggesting that hemocytes and some trachea-associated cell types are likely to have the same level of permissiveness to acquire viral particles circulating in the hemolymph. Immunologically active, sessile hemocytes are frequently associated with tracheas; this association has been suggested to put them at an immunological advantage because they are sites of gas exchange, with a high propensity for pathogen contact [53].

Our results suggest that both circulating and sessile hemocytes can rapidly sequester viral particles as they do with the nanobeads, potentially limiting virus dissemination to other tissues. However, tracheal structures displayed a comparable level of permissiveness to nanobeads as hemocytes, suggesting that viruses may invade tissues via tracheal entry, and that hemocytes and tracheae have a similar likelihood of becoming infected when viruses are circulating in the hemolymph.

## 3. Discussion

Hemocytes are the only cell type in mosquitoes that perform cellular immune responses such as phagocytosis, encapsulation, and nodulation [54,55]. They also produce most of the mosquito’s major immune factors, including PPOs, complement-like pathway proteins such as TEP1 and APL1C, and other humoral factors [13,14,56]. While our understanding of hemocyte immune responses against bacteria and malaria parasites [17,57,58,59,60,61,62,63,64] is extensive, the antiviral mechanisms of hemocytes have received less attention [11,16].

We show here that un-infected hemocytes exhibit high expression of RNAi pathway genes and that dengue virus infection upregulates additional genes associated with this pathway, suggesting that RNAi is an antiviral mechanism of hemocytes. Genes of the piRNA and miRNA pathways were the most abundantly expressed in hemocytes. miRNA pathway genes include Loqs, AGO1b, and Ars2, all of which have been shown to participate in the biogenesis of miRNAs in *Drosophila* [31,65,66]. Some of the highly expressed genes of the piRNA pathway include Zucchini, PIWI4, AGO3, and Atari, which participate in the *Ae. aegypti* antiviral immune response [24,33,67,68,69]. The DENV infection specifically upregulates PIWI5 of the piRNA pathway, as well as SAF-B2 and Cbp20 of the miRNA pathway. We also found that hemocytes of infected mosquitoes upregulate several transposable elements (TEs) that are known to be primarily controlled by the piRNA pathway [70]. Recent studies have shown that transposable elements (TEs) are crucial in generating endogenous viral elements (EVEs), which are sequences of non-retroviral viruses that become integrated into the mosquito genome and serve as templates for the production of viral piRNAs (vpiRNAs), a key component for controlling viral infections [32,33,71,72]. Although *Ae. aegypti* expresses eight different PIWI proteins (PIWI1-7 and AGO3), only PIWI4, PIWI5, and AGO3 have demonstrated antiviral functions, either generating piRNAs directly from viral RNA genome or acting through EVEs via poorly understood mechanisms [24,33,67,70]. Our transcriptomic analyses showed that hemocytes highly express several components of these pathways under basal conditions and upregulate additional components during viral infection, suggesting that they may mediate antiviral immune responses through these pathways.

Both insect hemocytes and fat body are the major immune-competent cells and tissue involved in pathogen elimination [54,73]. The transcriptomic analysis of the fat body and hemocytes revealed contrasting but complementary patterns of immune gene expression, suggesting a functional synergy necessary for triggering controlled and specific immune responses. (Figure 6). Many of the differentially expressed genes between the fat body and hemocytes are likely to function through the same pathway. For example, several CLIP-domain serine protease genes are highly expressed in the fat body, while PPOs are highly expressed in hemocytes. CLIP-domain serine proteases are known to activate melanization through the activation of PPOs along with other defense reactions [74,75,76]. Melanization occurs when a pathogen invades the mosquito; we hypothesize that expression of PPOs in hemocytes and CLIP-domain serine proteases in the fat body may result in better regulation and prevention of autoactivation. The VAGO protein has been suggested to act as a cytokine-like molecule that activates the JAK/STAT pathway and restricts West Nile virus (WNV) and DENV infection in mosquitoes [77,78,79]. We identified several intracellular components of the JAK/STAT pathway as being highly expressed in hemocytes, while VAGO1 and VAGO2 were expressed in the fat body. This differential expression pattern may also indicate inter-organ communication concerning the regulation of immune responses. It is plausible that compartmentalization of immune factors in different immune cells may allow mosquitoes to better regulate their immune responses (Figure 6a).

While the primary function of hemocytes is to kill pathogens, studies in *Drosophila* have shown that these cells play additional roles that indirectly influence the immune response, such as clotting, tissue morphogenesis and regeneration, which depend on their production of collagens, laminins, and other proteins [37,80,81]. Furthermore, we recently reported that hemocytes are key to maintaining the integrity of the midgut epithelium in *An. gambiae*; phagocyte-depleted mosquitoes show high mortality after blood feeding because of the resulting disruption of the midgut epithelium [39]. Interestingly, we found several collagen and laminin genes being highly expressed in hemocytes. However, phagocyte-depleted *Ae. aegypti* did not show any difference in mortality rate from that of control mosquitoes after blood feeding. Recent studies corroborate our findings, since they show that *Ae. aegypti* have a more robust intestinal epithelium and more effective tissue regeneration than do Anophelines [40,82,83]. Further studies are needed to better understand physiological and immunological differences in hemocytes between mosquito species and their impact on infection susceptibility.

Our transcriptomic analysis revealed that hemocytes exhibit higher transcriptional expression of the JNK, JAK/STAT, and IMD pathway genes than does the fat body. In *Ae. aegypti*, both the JNK and JAK/STAT pathways limit viral replication [78,84], and they play a major role in the interaction of sessile hemocytes with the heart of *An. gambiae* during an infection [85]. In *Drosophila*, the IMD pathway plays a key role in the antiviral immune response, which depends on hemocytes; over-expression of Relish in hemocytes suppresses systemic viral infection of Sindbis virus (SINV) [86,87]. Since we found a high expression of these genes in hemocytes of non-infected *Ae. aegypti* mosquitoes, this may enable the cells to more rapidly respond to viral infection.

Our results have shown that phagocytic hemocytes also contribute towards limiting systemic viral infections. Various studies have reported that when phagocytic cells are depleted, or phagocytosis is inhibited, higher viral loads are observed in mice, *Drosophila*, and mosquitoes [19,20,21,88,89], indicating a conserved antiviral function of phagocytosis and phagocytic cells. Our experiments showed that phagocytic hemocytes only control systemic DENV infection and not in the midgut. This suggests that phagocytes could prevent virus dissemination through the mosquito’s body, but once they reach other tissues, hemocytes most likely cannot modulate the infection into tissues.

As sentinel cells, hemocytes circulate and attach to tissues that are important for infection, increasing their chances of encountering pathogens [45,53,90]. This feature likely allows hemocytes to eliminate significant amounts of viral particles and thereby prevent dissemination to other tissues. We addressed this possibility experimentally by injecting fluorescent nanobeads with a size similar to that of virus particles, as a proxy to track the cells and tissues with physical permissiveness to DENV. Hemocytes indeed rapidly took up large numbers of nanobeads, as did the tracheas. Since several viruses, including DENV, SINV, o’nyong-nyong virus (ONNV), and Japanese encephalitis virus (JEV), can infect mosquito hemocytes [28,91,92,93,94], these cells have been suggested to facilitate their amplification and spread to other tissues such as the trachea [94,95]. However, most of these studies showed that hemocytes and the tracheas become infected in parallel [22,96], and during DENV infection tracheas is the first tissue to become infected [28]. This suggests that the virus disseminates from the midgut and infects other tissues at early times, and no previous viral amplification is required. This may explain the detection of CHIKV and DENV in the salivary gland and carcass tissues of *Ae. aegypti* already at 2 days post-ingestion of infected blood [26,28,97]. Furthermore, hemolymph flow is also likely to transport virus particles throughout the mosquito body as it does for nutrients, hemocytes, and pathogens such as *Plasmodium* sporozoites which by this way reach salivary glands [45,46,90,98].

It is important to note that our study focused on understanding the role of hemocytes during the early stages of DENV infections, specifically when the virus infiltrates the hemolymph and begins to spread. While our results indicate that hemocytes play a role in controlling systemic viral infection, we cannot exclude the possibility that they may also influence viral dynamics in other tissues or act as agonists at different stages of infection, as previously reported. [19,95]. The limitations in understanding the molecular mechanisms underlying the antiviral immune responses mediated by hemocytes is largely due to the absence of advanced genetic tools in this mosquito model, which prevents hemocyte-specific manipulation of gene expression, unlike what is feasible in *Drosophila* [99,100].

In summary, our study presents for the first time a transcriptomic comparison between *Ae. aegypti* hemocytes and fat body, and between DENV-infected and naïve hemocytes. The transcriptomic analyses suggested that hemocytes and the fat body play complementary and synergistic roles in immune defense. Our findings also suggest that hemocytes play an important role in controlling viral infections in *Ae aegypti*, specifically at the level of systemic DENV infection by internalizing viruses present in the hemolymph and then eliminating them via the piRNA pathway (Figure 6b). Overall, our study provides evidence supporting the involvement of hemocytes in the antiviral immune response.

## 4. Materials and Methods

### 4.1. Mosquitoes Rearing

*Ae. aegypti* Liverpool strain mosquitoes were maintained at 80% relative humidity, 27 °C, with a 14 h:10 h light/dark cycle. Larvae were fed on fish food pellets (TetraMin tropical tablets, 1.69 oz). Adult mosquitoes were fed on 10% sucrose solution. Mosquito rearing and experiments were performed in the Insectary Core Facility of the Johns Malaria Research Institute.

### 4.2. Cell Culture and Virus Propagation

*Ae. albopictus* C6/36 cells (ATCC, Manassas, VA, USA; SKU: CRL-1660) were cultured in Minimum Essential Medium (MEM (Gibco, Waltham, MA, USA; SKU:12360-038)) supplemented with 10% fetal bovine serum (FBS (Sigma, St. Louis, MO, US; SKU: S8761)), 1% penicillin (10,000 U/mL)/streptomycin (10,000 μg/mL); (Gibco, SKU: 15140-122), 0.2% Plasmocin (InvivoGene, San Diego, CA, USA; SKU: ant-mpp), 1% L-glutamine (200 mM; Gibco, SKU: 25030-081), 1% MEM non-essential amino acids 100x (Quality Biological, Gaithersburg, MD, USA; SKU: 116-078-721), at 32 °C in 5% CO_2_.

For virus propagation, C6/36 cells were cultured to 50% confluence in MEM (supplemented as usual, but with only 2% FBS) and infected by adding 1 mL of dengue virus serotype 2 (New Guinea C Strain) stock (1 × 10^8^ PFU/mL). The cells then were incubated for 5 days, and virus was then harvested as described by Das et al., 2007 [101].

Baby hamster Kidney cells (BHK-21 (ATCC, Manassas, US; SKU: CCL-10)) were cultured in Dulbecco’s Modified Eagle Medium (DMEM (Gibco, SKU: 11965-092)) supplemented with 10% FBS (Sigma, SKU: S8761), 1% penicillin (10,000 U/mL)/streptomycin (10,000 μg/mL); (Gibco, SKU: 15140-122), and 0.2% Plasmocin (InvivoGene, SKU: ant-mpp).

### 4.3. Viral Infections and Titrations

Midgut infections were performed by feeding 3-day-old female *Ae. aegypti* mosquitoes on reconstituted blood containing DENV (45% red blood cells; 40% DENV viral stock, 1 × 10^7^ pfu/mL; 10% human serum). Mosquitoes were dissected at 7 days post-infection and each midgut was collected in 300 μL of DMEM infection medium (supplemented as usual, but with only 2% of FBS) containing 100 μL of 0.5 mm-diameter glass beads (Next Advance, Inc.; New York, NY, USA). For systemic infections, mosquitoes were injected with 69 nL of MEM medium containing 10,000 pfu/μL of DENV. At 7 days post-injection, each whole mosquito body was collected in 300 μL of DMEM infection medium containing 100 μL of 0.5 mm-diameter glass beads. Samples were stored at −80 °C until processed.

Infections were evaluated by plaque assay according to a previously described method [102]. Samples of whole mosquitoes and tissues were homogenized for 5 min at speed 9 in a Bullet Blender 24 (Next Advance, Inc.). Serial logarithmic dilutions of 100 μL were placed in 24-well plates containing BHK-21 cells (ATTC, SKU: CCL-10) at 50% confluence. Each well was then filled with 900 μL of DMEM infection medium containing 0.8% methyl cellulose (Sigma, SKU: M0512). The plates were incubated for 7 days at 37 °C with 5% CO_2_ and then stained with 1% crystal violet solution (in 1:1 methanol/acetone) to visualize the plaques.

### 4.4. Phagocytic Hemocyte Depletion

Mosquito phagocyte depletion was performed according to a previously described method, with some modifications [34]: female mosquitoes (3 days old) were injected intra-abdominally with 69 nL of a 1:5 suspension in RPMI 1640 medium (Gibco, SKU: 21870076) of either control liposomes (LPSMs) or clodrosomes (CLDs) (Standard Macrophage Depletion Kit, clodrosome + encapsome; SKU: CLD-8901-2 mL, Encapsula Nanosciences). All experiments were performed at 4 days post-injection.

### 4.5. Hemolymph Perfusion and Hemocyte Staining

To recover circulating hemocytes, we followed a protocol previously described [38]. Mosquito females were injected with 5 μL of perfusion buffer (60% Schneiders’s medium [Gibco, SKU: 21720-024], 10% fetal bovine serum (Sigma, SKU: 22L538), and 30% citrate buffer [98 mM NaOH, 186 mM NaCl, 1.7 mM EDTA, and 41 mM citric acid, at pH 4.5]) containing Vybrant CM-DiI Cell-Labeling Solution (5 μL/mL; Invitrogen, SKU: V22888). The injected mosquitoes were kept in ice for 30 min and perfused by cleaving the membrane of the last segment and injecting 5 μL of perfusion buffer into the thorax. The perfused hemolymph was collected on a glass slide and incubated in a humid chamber for 30 min at room temperature. The cells that attached to the glass slide were fixed by adding 20 μL of 4% of formaldehyde in Phosphate-Buffered Saline (PBS). The cells were then incubated for 10 min at room temperature in a humid chamber and washed three times with fresh PBS for 10 min each. The cells were mounted in 10 μL Fluoromount G with DAPI (Invitrogen SKU: 00-4959-52) and coverslipped.

### 4.6. RNAseq: Tissue Collection, RNA Isolation, Library Preparation, and Analysis

To collect circulating hemocyte samples, we followed a previously described protocol [85]; 150 female mosquitoes (DENV-infected and non-infected) were perfused at 10 days post-infection. The perfusion was performed by cleaving the membrane of the last abdominal segment and injecting Diethyl pyrocarbonate (DEPC)-treated PBS into the thorax. To obtain the fat body cells, we followed a protocol previously used to recover fat body tissue from the mosquito heart [103] with some modifications. Following perfusion, the abdomens were separated from the thorax in a drop of DEPC-treated PBS. The lateral membrane of each abdomen was cut using a 20G1 syringe needle blade (BD) and an entomological needle (size 0). The syringe needle blade was inserted into the abdominal hemocoel along the lateral membrane, and the entomological needle was used to carefully scratch along the membrane over the needle blade until the abdomen was completely opened. The opened abdomens, with the hemocoel exposed, were then gently scraped with the entomological needle to recover primarily fat body cells. The perfused hemolymph and fat body cells were collected directly into RTL Buffer (from the RNeasy Micro Kit) supplemented with 2% 2-mercaptoethanol, and isolation was performed according to the instructions of the RNeasy Micro Kit (Qiagen, Hilden, Germany; SKU: 74004).

Library construction, sequencing, and analysis were performed by NOVOGEN Co., LTD (Beijing, China). The mRNA was purified from 100 ng of total RNA using poly-T oligo attached to magnetic beads. After fragmentation, the first strand of cDNA was synthesized using random hexamer primers, followed by second-strand cDNA synthesis using dUTP to produce a directional library or dTTP to produce a non-directional library. The non-directional library was prepared by end repair, A-tailing, adapter ligation, size selection, amplification, and purification. For directional library preparation, end repair, A-tailing, adapter ligation, size selection, USER enzyme digestion, amplification, and purification were performed. The libraries were checked with Qubit and real-time PCR for quantification and were bioanalyzed to analyze size distribution. Quantified libraries were sequenced on a NovaSeq6000 sequencing system (Illumina) with paired ends of 150 bp. The FastQ format contains the clean data (i.e., reads without adapters or low-quality reads from raw data). All the downstream analyses were based on the clean data with high quality.

The paired-end clean reads were aligned with the reference genome (VectorBase_57_Aaegypti) using Hisat2 v2.0.5. The mapped reads of each sample were assembled by StringTie (v1.3.3b) in a reference-based approach. To count the read number mapped for each gene, featureCounts v1.5.0-p3 was used. The fragments per kilobase of transcript sequence per million base pairs sequenced (FPKM) was calculated based on the length of the gene and reads counts mapped to that gene. FPKM takes into account both the sequencing depth and the length of the gene when calculating read counts. It is currently the most widely used method for estimating gene expression levels. Differential gene expression analysis of two conditions/groups was performed with the DESeq2R package (1.20.0) that provides statistical routines to determine differential expression using a negative binomial distribution model. The resulting *p*-values were adjusted using Benjamini and Hochberg’s test, and *p*-values ≤ 0.05 were assigned as differentially expressed. Differentially expressed genes were enriched in GO terms and KEEG pathways by using the clusterProfiler R package (Bioconductor 3.15).

### 4.7. Fluorescent Nanobeads Assays and Microscopy

Three-day-old female mosquitoes were injected in the abdomen with 69 nL of RPMI 1640 (Gibco, SKU: 21870076) containing 10% of 30 nm carboxylate-modified polystyrene fluorescent yellow–green latex beads (Sigma, SKU: L5155). Mosquitoes were incubated at 28 °C, and then at 2, 4, and 24 h, then groups of five mosquitoes each were injected in the abdomen with 250 nL of RPMI 1640 containing the hemocyte marker dye Vybrant CM-DiI Cell-Labeling Solution (Invitrogen, SKU: V22888) [5 μL/mL]. Mosquitoes were incubated at 4 °C for 20 min, then dissected in a drop of PBS, and the ovaries and midgut were removed from the abdomen. The abdominal carcass was separated from the thorax and opened by cutting the lateral membrane. All the tissues were washed three times with PBS. The ovaries, midgut, and abdomen were incubated in 4% formaldehyde in PBS for 30 min, followed by three washes with PBS for 10 min each at room temperature. The samples were placed in a drop of Fluoromount G with DAPI (Invitrogen SKU: 00-4959-52) on a glass slide and coverslip. In parallel, hemocytes were obtained according to the standard protocol described in the previous section (“Hemolymph perfusion and hemocyte staining”).

Images were obtained using a LEICA DM2500 microscope with a LEICA K3C camera and analyzed using Fiji software, version 2.14.0/154f [104].

### 4.8. Smurf Assay

Midgut epithelial integrity was analyzed by Smurf assay as previously described [39]. Mosquitoes were fed on reconstituted blood (50% RBC, 48% human blood serum, 1% 100 mM ATP) containing a 1% Smurf solution (25 g/L of FD&C blue dye #1 [Sigma, SKU: 861146]) in buffered saline solution (150 mM NaCl and 10 mM NaHCO_3_, pH 7.5). The Smurf phenotype (blue mosquitoes) and mortality were evaluated 48 h after blood feeding.

### 4.9. Statistical Analysis

All data were generated from at least three biological replicates; raw data are available in the Appendix A file. All data were analyzed using an appropriate statistical test in GraphPad Prism V10.4.1. The statistical test for each experiment is described in the corresponding figure legend, and more detailed information is available in the Appendix A file.

## Figures and Tables

**Figure 1 ijms-26-08779-f001:**
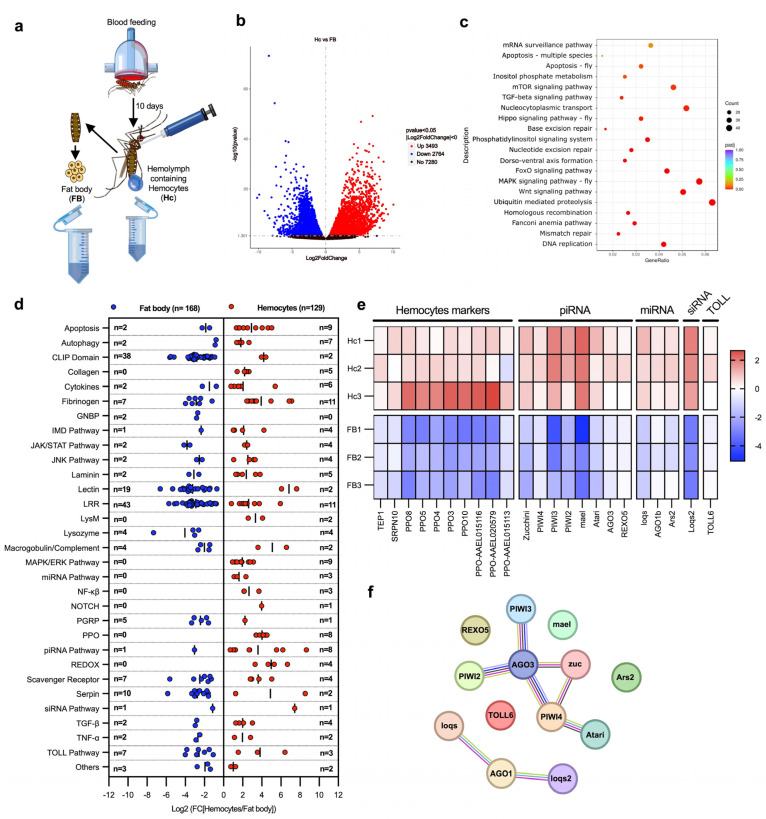
Transcriptomic analysis of immunity-related genes in the fat body and hemocytes. (**a**) Experimental design for comparing fat body and hemocyte transcriptomic profiles. Fat body cells and hemolymph from 150 female mosquitoes (per replicate) were collected at 10 days post-blood feeding to analyze their transcriptomes. (**b**) Volcano plot of differentially expressed genes on hemocytes (*p* < 0.05). Dots in blue indicate downregulated genes in hemocytes (which are highly expressed in the fat body); dots in red show upregulated genes in hemocytes; black dots indicate non-differentially expressed genes. The *X*-axis indicates the Log2 of fold change, and the *Y*-axis indicates the -Log10 of *p*-value. (**c**) Scatter plot of KEEG pathway enrichment of highly expressed genes in hemocytes. The circle area indicates the number of upregulated genes in hemocytes; the color scale indicates the adjusted *p*-value. *X*-axis indicates gene ratio (count of core enrichment genes/count of pathway genes). (**d**) Scatter plot of immune genes classified by pathways and families. Blue dots indicate highly enriched gene mRNAs in the fat body (with lower expression in hemocytes); red dots denote highly expressed gene mRNAs in hemocytes (lower expression in the fat body). The *X*-axis indicates the Log2 fold change [Hemocytes/Fat body]. The number of genes is indicated on *Y*-axis as “n =”. (**e**) Heat map of Z-score values of hemocyte marker genes and antiviral immune genes. The red color indicates upregulated genes in hemocytes; the blue color shows downregulated genes. The heat map presents the values of each replicate of hemocytes (Hc1-3) and fat body (FB1-3) samples. (**f**) String protein–protein interaction analysis of antiviral genes highly expressed in hemocytes. Each circle represents a protein, and each line represents an interaction (physical or functional). All data were obtained from three biological replicates. Differentially expressed genes were determinate using DESeq2 package; the *p*-values were adjusted by Benjamini and Hochberg’s test, *p* ≤ 0.05 were assigned as differentially expressed. (**a**) was created by Victor Cardoso-Jaime in Microsoft PowerPoint Version 16.92.

**Figure 2 ijms-26-08779-f002:**
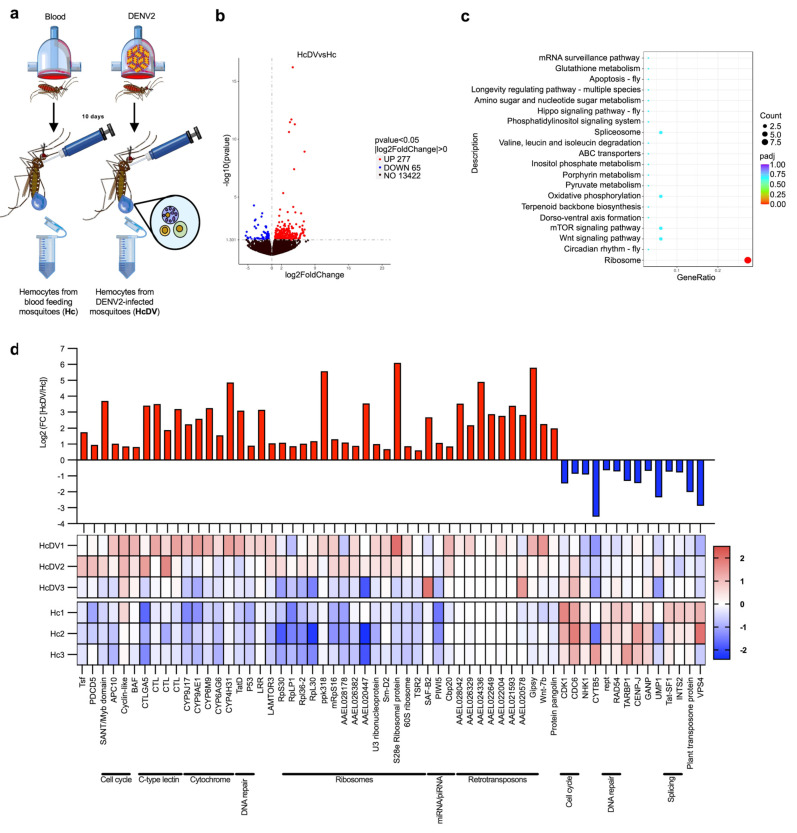
Transcriptomic analysis of hemocytes from dengue virus-infected mosquitoes. (**a**) Experimental strategy for transcriptomic analysis of hemocytes. Comparison of transcriptomic profiles in hemocytes from DENV-infected mosquitoes was performed 10 days post-infection, using hemocytes from blood-fed mosquitoes as controls. (**b**) Volcano plot of differentially expressed genes in hemocytes from DENV-infected mosquitoes. The blue dots represent genes that are downregulated, and red dots denote genes that are upregulated in the hemocytes of DENV-infected mosquitoes. The *X*-axis indicates the Log2 of fold change, and the *Y*-axis indicates the -Log10 of *p*-value. (**c**) Scatter plot of KEEG pathway enrichment of upregulated genes in hemocytes of DENV-infected mosquitoes. The circle areas indicate the number of upregulated genes; the color scale indicates the adjusted *p*-value; *X*-axis indicates gene ratio (count of core enrichment genes/count of pathway genes). (**d**) Log2 of -fold change values, and z-score heat map of differentially expressed genes in the hemocytes of DENV-infected mosquitoes. The red color indicates upregulated genes in hemocytes, and the blue color represents downregulated genes. The heat map presents the values of each replicate of hemocytes from DENV-infected (HcDV1-3) and naive (Hc1-3) mosquitoes. All data were obtained from three biological replicates. Each experiment was performed using samples from 150 female mosquitoes. Differentially expressed genes were determined using DESeq2 package; the *p*-values were adjusted by Benjamini and Hochberg’s test, *p* ≤ 0.05 were assigned as differentially expressed. (**a**) was created by Victor Cardoso-Jaime in Microsoft PowerPoint Version 16.92.

**Figure 3 ijms-26-08779-f003:**
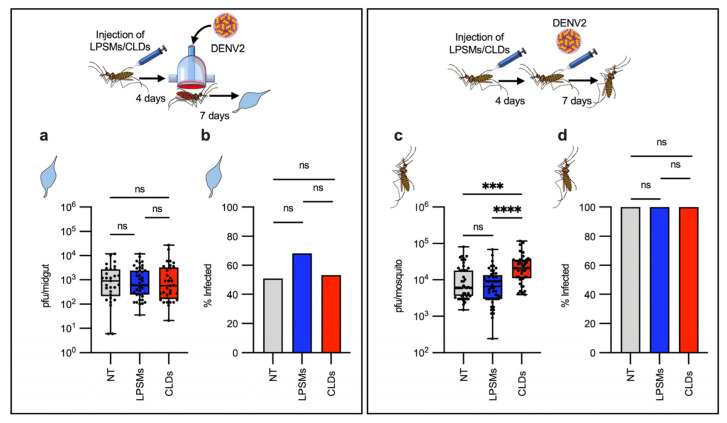
Effect of phagocytic hemocyte depletion in dengue virus infection. (**a**,**b**) Non-treated [NT], control liposome-injected [LPSMs], and clodrosome-injected [CLDs] mosquitoes were fed on blood containing DENV, and then the viral load (**a**) and prevalence (**b**) were evaluated in the midgut at 7 days post-infection. (**a**,**b**) n= 57 (NT), n= 63 (LPSMs), n= 62 (CLDs). (**c**,**d**) NT, LPSMs, and CLDs mosquitoes were infected by directly injecting the same number of viral particles into the hemocoel. Viral titer (**c**) and prevalence (**d**) were then evaluated in the whole mosquitoes at 7 days post-infection. (**c**,**d**) n= 48 (NT), n = 48 (LPSMs), n = 48 (CLDs). Viral titers are expressed as a median with a min. to max. of plaque-forming units (PFUs), represented in a box and whiskers graph, where each dot represents an individual sample (midgut or whole mosquito). Statistical differences in viral titers between groups were analyzed using a Kruskal–Wallis test, followed by a Dunn’s multiple comparation test. Prevalence values are expressed as percentages in a bar graph. Statistical analysis of prevalence was performed using a two-sided Fisher’s exact test. *** *p* = 0.0001; **** *p* < 0.0001; ns, not significant. Experimental strategy diagrams were created by Victor Cardoso-Jaime in Microsoft PowerPoint Version 16.92.

**Figure 4 ijms-26-08779-f004:**
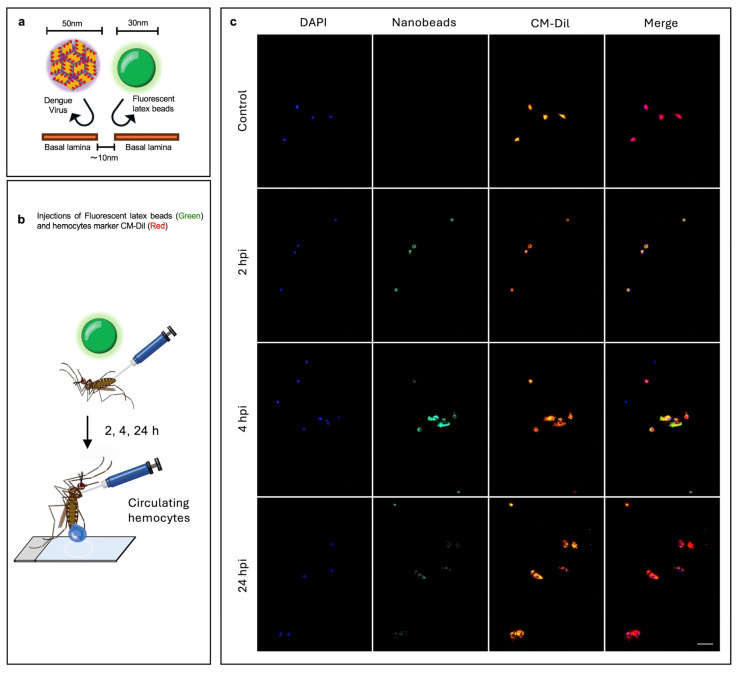
Analysis of hemocyte capacity to take up nanobeads. (**a**) Experimental strategy: the basal lamina of tissues are permeable to particles smaller than 10 nm, while DENV and nanobeads are bigger than 10 nm, meaning that the tissues cannot take up nanobeads and viruses, except in the case of cells lacking a basal lamina, such as hemocytes. (**b**) Mosquitoes were injected with fluorescent nanobeads 30 nm sized, and then hemolymph was perfused to collect the hemocytes at 2, 4, and 24 h post-injection. (**c**) Images at 2, 4, and 24 h post-injection of hemocytes from mosquitoes injected with nanobeads; hemocytes from mosquitoes injected with the vehicle were used as controls. In blue, the nuclei of cells stained with DAPI; green, 30 nm fluorescent nanobeads; red, CM-Dil (a hemocyte-specific dye); scale bar = 20 μm. Images are representative of three biological replicates (original images (Raw Images) of replicates, Appendix A). Experimental strategy diagrams were created by Victor Cardoso-Jaime in Microsoft PowerPoint Version 16.92.

**Figure 5 ijms-26-08779-f005:**
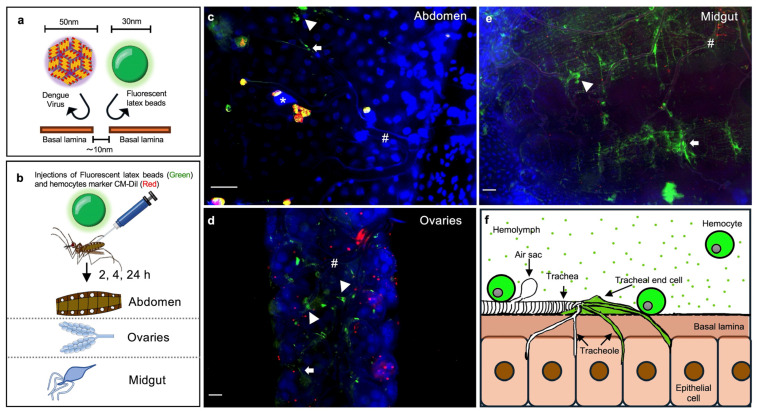
Permeability of mosquito tissues to virus-sized nanobeads. (**a**) Experimental strategy for assessing tissue permeability to viruses. Epithelial tissues are surrounded by basal lamina permeable to particles smaller than 10 nm; the diameter of DENV is 50 nm, and the size of the similarly sized, virus-mimicking fluorescent nanobeads used is 30 nm. (**b**) Mosquitoes were injected with nanobeads, and next tissues were dissected at 2, 4, and 24 h post-injection (all time points are shown in Appendix A). (**c**–**e**) Representative images of the distribution of fluorescent nanobeads and hemocytes in the abdomen (**c**), ovaries (**d**), and midgut (**e**) at 4 h post-injection. Blue, nuclei stained with DAPI; red, hemocytes stained with CM-Dil; green, 30 nm fluorescent nanobeads; *, air sacs; #, trachea; arrowhead, tracheal end cells; arrows, tracheoles; bar = 25 μm. (**f**) Model depicting the permeability of epithelial tissues to virus-sized nanobeads. Nanobeads are absorbed by the tracheal end cells, which are located at the ends of the tracheal branches where the tracheoles converge. Circulating and sessile hemocytes also acquire nanobeads. Images are representative of three independent biological replicates (original Images (Raw Images) of replicates, Appendix A). (**a**,**b**,**f**) were created by Victor Cardoso-Jaime in Microsoft PowerPoint Version 16.92.

**Figure 6 ijms-26-08779-f006:**
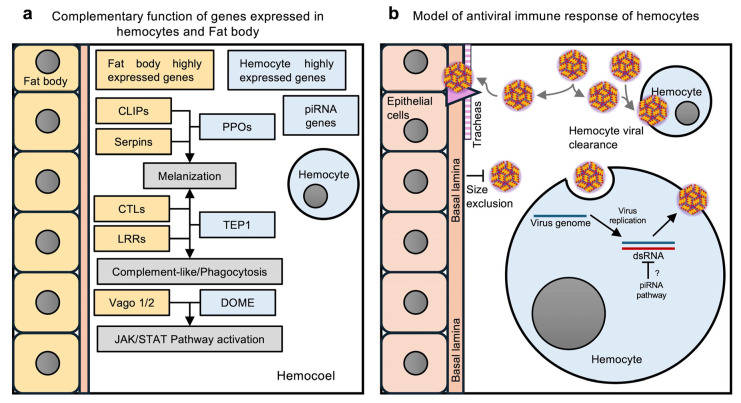
Immune responses by *Ae. aegypti* hemocytes. (**a**) Hypothetical model of complementary functions of immunity-related genes expressed in the fat body and hemocytes. The fat body expresses the genes encoding CLIP-domain proteins (CLIPs) and serpins, which regulate the phenoloxidase (PO) cascade. However, hemocytes produce the precursors of POs, the prophenoloxidases (PPOs). The melanization of pathogens results from the combined function of these genes. The CTLs and LRR genes expressed in the fat body work with the TEP1 produced by hemocytes to activate melanization, the complement-like pathway, or phagocytosis. The fat body expresses the Vago1/2 cytokine-like proteins of the JAK/STAT pathway, and hemocytes highly express several other genes of the JAK/STAT pathway, including DOME, the putative receptor of Vago. Activation of the JAK/STAT pathway protects mosquitoes from viral infections. (**b**) Hypothetical model of hemocyte antiviral immune responses. Viruses circulating in the hemolymph infect only the hemocytes and tracheae. The tracheae serve as the route of infection for epithelial cells, since epithelial tissues are surrounded by a basal lamina that prevents virus entry by size exclusion. Hemocytes control virus dissemination through viral clearance. When hemocytes acquire the virus, they manage viral infection through the piRNA pathway. (**a**,**b**) were created by Victor Cardoso-Jaime in Microsoft PowerPoint Version 16.92.

## Data Availability

The raw data for each graph are provided along with this article as the Appendix A file. The RNAseq data generated in the current study were deposited in the NCBI’s Sequence Read Archive (SRA) under accession number PRJNA1214865.

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
