# Peer review of "Aedes aegypti Hemocytes Mediate Antiviral Immunity"

_ijms, 2025, doi:10.3390/ijms26188779_

Round 1

Reviewer 1 Report

Comments and Suggestions for Authors

This manuscript about antiviral immunity in Aedes aegypti is reporting an interesting transcriptomic study about the role of hemocytes and fat body in the immune response in Aedes aegypti against dengue virus. The authors elegantly show the importance of the phagocytic activity of hemocytes to restrict spreading of the virus throughout the body. This is an excellent study and should be of great interest to insect immunologists.

There are only two minor comments:

Figure 1e: explain what you man by Hc1, Hc2, Hc3, Fb1, Fb2, Fb3 in the legend. Are these from individual animals, then comment the big variation in PPO transcripts (Hc3). (in general the figure legends could be a bit more detailed).

Supplementary table 6: format column for gene ratio as text, since now it shows dates (i.e. feb-33 …)

Author Response

We appreciate the Reviewer comments, which we have addressed by adding detailed information on figure legends and correcting the supplementary material. The corresponding corrections are highlighted in red to track changes in the resubmitted files.

Comments and Suggestions for Authors

This manuscript about antiviral immunity in Aedes aegypti is reporting an interesting transcriptomic study about the role of hemocytes and fat body in the immune response in Aedes aegypti against dengue virus. The authors elegantly show the importance of the phagocytic activity of hemocytes to restrict spreading of the virus throughout the body. This is an excellent study and should be of great interest to insect immunologists.

There are only two minor comments:

Comments 1: Figure 1e: explain what you man by Hc1, Hc2, Hc3, Fb1, Fb2, Fb3 in the legend. Are these from individual animals, then comment the big variation in PPO transcripts (Hc3). (in general the figure legends could be a bit more detailed).

Response 1: We have added more details to all figure legends, including the number of animals used per experiment. For the transcriptome analyses, we used 150 female mosquitoes to obtain hemolymph and fat body samples for each replicate. In each experiment, hemolymph and fat body samples were obtained from the same animals. The variation between samples is likely due to each replicate being performed with a different mosquito batch. To ensure robustness, we only considered genes as differentially expressed if they had a p-value < 0.05, and we only show genes with statistically significant differences (Figure 1-2 legends, lines. 97-119, 158-175).

Comments 2: Supplementary table 6: format column for gene ratio as text, since now it shows dates (i.e. feb-33 …)

Response 2: We have made these changes. Now Supplementary table 6 presents ratios (n/1933).

Reviewer 2 Report

Comments and Suggestions for Authors

The study "Aedes aegypti Hemocytes Mediate Antiviral Immunity" conducted a comparative transcriptomic analysis of the dengue vector's primary immune tissues, hemocytes and fat body. This analysis revealed extensive differential expression of immune genes, highlighting significant functional specialization and complementarity between the two cell types. Notably, hemocytes exhibited high basal expression of RNAi genes and upregulated RNAi pathway components following dengue virus infection. The manuscript presents convincing data and is well-written.

Minor issues:

The resolution of Fig1c image needs to be increased

Author Response

We appreciate the comments of Reviewer 2 that helped to improve the quality of your manuscript. 

Comments and Suggestions for Authors

The study "Aedes aegypti Hemocytes Mediate Antiviral Immunity" conducted a comparative transcriptomic analysis of the dengue vector's primary immune tissues, hemocytes and fat body. This analysis revealed extensive differential expression of immune genes, highlighting significant functional specialization and complementarity between the two cell types. Notably, hemocytes exhibited high basal expression of RNAi genes and upregulated RNAi pathway components following dengue virus infection. The manuscript presents convincing data and is well-written.

Minor issues:

Comments 1: The resolution of Fig1c image needs to be increased

Response 1: We have increased the resolution of Fig. 1c and we have added the original figure with this new submission.

Reviewer 3 Report

Comments and Suggestions for Authors

The manuscript titled "Aedes aegypti Hemocytes Mediate Antiviral Immunity" conducted a transcriptomic analysis on the two primary immune cell types of Aedes aegypti, namely hemocytes and fat body. The study involved infecting hemocytes with dengue virus and identifying immune genes that exhibited differential expression. It confirmed the role of phagocytic hemocytes in controlling systemic infection and suggested the ability of hemocytes and tracheas to internalize viral-sized nanobeads. These findings contribute to a better understanding of Dengue virus infection within hemocytes in Aedes aegypti. 

However, several details and conclusions require revision for a more accurate interpretation of the manuscript with solid data support.

1. In the abstract, the statement "hemocytes control viral infection most likely through the RNAi pathways" should be revised as the current data does not sufficiently support this conclusion or hypothesis.

2. The introduction should briefly introduce arthropod-borne viruses transmitted by mosquitoes before focusing on Dengue virus to provide context.

3. The second paragraph and third paragraph are repetitive. To avoid redundancy, the content about the cellular immune response should be integrated into a single coherent section.

4. When abbreviations such as TEP1, APL1C, FBN9, CLIP, etc., first appear in the text, they should be clearly explained to ensure clarity.

5. The claim that highly expressing genes of the hemocytes are major RNAi antiviral pathways (miRNA, siRNA, and piRNA) (Fig. 1c) needs revision because according to Fig. 1c, both enrichment genes in miRNA and siRNA pathway have only one count (n=1).

6, For Fig. 1d, it is recommended to switch hemocytes to the left side and fat body to the right side to align with the sequence of the text for better readability.

7. In the figure captions and statistical analysis, please specify the sample size for each biological replicate to enhance reproducibility and transparency.

8. Regarding the figures, while it states "Image are representative of three biological replicates," it would be beneficial to provide the other two replicate images in the supplementary figures for a more comprehensive view.

9. In the discussion, the statement "suggesting that RNAi serves as the primary antiviral mechanism in hemocytes" should be revised since the current data does not strongly support this suggestion.

10. To validate that RNAi pathway serves as the primary antiviral mechanism in hemocytes, additional experimental validation is required.

11. Based on the concerns raised, the model depicted in Fig. 6b is not yet conclusively proven and should be cautiously interpreted until further evidence is provided.

12. In Materials and Methods (M&M), terms like FBS, BHK, DMEM should be defined or abbreviated consistently throughout the text for clarity.

Author Response

We appreciate the reviewer comments, which have helped us improve the quality of our manuscript. We have addressed these comments by adding detailed information to the figure legends and providing new supplementary material, and rewriting the conclusions. The corresponding corrections are highlighted in red to make it easy to track changes in the resubmitted files.

Comments and Suggestions for Authors

The manuscript titled "Aedes aegypti Hemocytes Mediate Antiviral Immunity" conducted a transcriptomic analysis on the two primary immune cell types of Aedes aegypti, namely hemocytes and fat body. The study involved infecting hemocytes with dengue virus and identifying immune genes that exhibited differential expression. It confirmed the role of phagocytic hemocytes in controlling systemic infection and suggested the ability of hemocytes and tracheas to internalize viral-sized nanobeads. These findings contribute to a better understanding of Dengue virus infection within hemocytes in Aedes aegypti. 

However, several details and conclusions require revision for a more accurate interpretation of the manuscript with solid data support.

Comments 1: In the abstract, the statement "hemocytes control viral infection most likely through the RNAi pathways" should be revised as the current data does not sufficiently support this conclusion or hypothesis.

Response 1: We modified the sentence: Our results suggest that hemocytes possess mechanisms to control systemic viral infections. (Lines 21-22).

Comments 2: The introduction should briefly introduce arthropod-borne viruses transmitted by mosquitoes before focusing on Dengue virus to provide context.

Response 2 : We have added a new sentence prior to discussing dengue, and we have also included a new reference:

“The risk of a pandemic resulting from an emerging or re-emerging arbovirus, such as dengue, Zika, yellow fever, and chikungunya viruses, is expected to continue increasing due to human movement, urbanization, climate change, and mosquitoes' adaptation have resulted in an uncontrolled expansion of the vectors [1].”. (Lines 28-31)

1. WHO Global Arbovirus Initiative: Preparing for the next Pandemic by Tackling Mosquito-Borne Viruses with Epidemic and Pandemic Potential 13 May 2024; WHO: Geneva, 2024; ISBN electronic version.

Comments 3: The second paragraph and third paragraph are repetitive. To avoid redundancy, the content about the cellular immune response should be integrated into a single coherent section.

Response 3:  We have modified the second and third paragraphs of the introduction, focusing on the innate immune response of mosquitoes (humoral and RNAi pathways) (Lines 37-49) and the cellular immune response (Lines 50-60), respectively.

Comments 4: When abbreviations such as TEP1, APL1C, FBN9, CLIP, etc., first appear in the text, they should be clearly explained to ensure clarity.

Response 4:  Now we explain the abbreviation when it is first mentioned.

Comments 5: The claim that highly expressing genes of the hemocytes are major RNAi antiviral pathways (miRNA, siRNA, and piRNA) (Fig. 1c) needs revision because according to Fig. 1c, both enrichment genes in miRNA and siRNA pathway have only one count (n=1).

Response 5: We have changed the sentence to:

“ The gene set enrichment analysis showed that hemocytes were highly expressing genes involved in the dorso-ventral axis formation pathway, a category that also includes genes of the major piRNA pathway (Figure 1c, and Supple- mentary Table 3)” Lines 80-83)

We are right that miRNA and siRNA pathways only have 3 and 1 genes highly expressed in hemocytes, respectively; however, when we say RNAi pathways, we refer to the piRNA, miRNA, and siRNA pathways. To avoid this confusion, we also changed the following sentence:

“As expected, PPO genes were also highly expressed in hemocytes. Interestingly, some of the most representative categories of highly expressed genes in hemocytes were components of the autophagy, and RNAi pathways (piRNA, miRNA, and siRNA pathways)” (Lines 88-91)

Comments 6: For Fig. 1d, it is recommended to switch hemocytes to the left side and fat body to the right side to align with the sequence of the text for better readability.

Response 6: The figure was designed according to the standard mathematical rules in which the right and upper axes represent positive values, while the left and lower axes represent negative values. Since we show genes that are higher expressed in hemocytes and, consequently, lower expressed in the fat body, we assigned the right and upper parts of the graphs to hemocytes, and the left and lower parts to the fat body.

Comments 7: In the figure captions and statistical analysis, please specify the sample size for each biological replicate to enhance reproducibility and transparency.

Response 7: We have added sample sizes and statistical methods in all figure legends. Additionally, we have provided all raw data and statistical analysis in the supplementary table “Source Data”.

Comments 8: Regarding the figures, while it states "Image are representative of three biological replicates," it would be beneficial to provide the other two replicate images in the supplementary figures for a more comprehensive view.

Response 8:  We have added this information to the supplementary material, please find them on the Supplementary material at Folder: Original Images (Raw images)>Replicates Figure 4, Supp.Figure 3-5. We also indicate this information at Figure legends of Figures 4 and 5.

Comments 9: In the discussion, the statement "suggesting that RNAi serves as the primary antiviral mechanism in hemocytes" should be revised since the current data does not strongly support this suggestion.

Response 9: We agree with the comment we have changed the statement by “suggesting that RNAi is an antiviral mechanism of hemocytes”. (Line 300)

Comments 10: To validate that RNAi pathway serves as the primary antiviral mechanism in hemocytes, additional experimental validation is required.

Response 10: We hypothesize that the RNAi pathways could serve as the primary antiviral mechanisms in hemocytes since they are overall crucial for antiviral defense in mosquitoes. Although we believe this hypothesis should be addressed, our current genetic tools in mosquitoes do not allow us to manipulate gene expression in a tissue-specific manner, which limits experiments exploring the function of specific cell types, such as hemocytes. Future studies will help to unveil these mechanisms. (Lines 413-417)

Comments 11: Based on the concerns raised, the model depicted in Fig. 6b is not yet conclusively proven and should be cautiously interpreted until further evidence is provided.

Response 11: We agree that we cannot completely support that the piRNA pathway is the primary antiviral pathway in mosquitoes. However, we have hypostatized that it can contribute towards controlling viral infection in hemocytes. We have changed the sentence in the Figure 6a-b legend to “hypothetical model”.

Comments 12: In Materials and Methods (M&M), terms like FBS, BHK, DMEM should be defined or abbreviated consistently throughout the text for clarity.

Response 12: We acknowledge the comment and have carefully ensured that we explain each abbreviation the first time it appears.

Reviewer 4 Report

Comments and Suggestions for Authors

This manuscript contains an RNAseq dataset that suggests that hemocytes and fat body play complementary roles in the immune response. Moreover, the authors use RNAseq and a chemical method to ablate hemocytes to demonstrate that hemocytes are important in controlling viral infections in the hemocoel but not in the midgut. This is an interesting manuscript that advances our understanding of the antiviral response in mosquitoes, and particularly, the antiviral response in the hemocoel. In that sense, this manuscript presents a significant advance in the field. Below I raise some concerns and make some suggestions. The authors should be able to easily respond to these comments.

  1. More detail is needed on tissue collection, and more specifically, how the fat body was collected. Depending on the method, some of the results and conclusions may need to be modified. The methods section does not detail how the fat body was collected, and a detailed protocol must be provided. Figure 1 implies that fat body cells were squeezed out of the abdomen, which I find unlikely. If the entire abdomen was used, then the samples would also contain abdominal muscle, ventral nerve cord, pericardial cells, heart, and other tissues (including sessile hemocytes). It would even contain circulating hemocytes if not perfused beforehand. If “fat body” is really the abdomen, can the percentage of the tissue that is fat body be calculated? How can the investigators be sure that their findings are attributable to fat body and not other cells. I am fairly confident that fat body is overabundant on the abdominal wall, but more detail is needed.
  2. I am excited about the potential that hemocytes and fat body each specialize on specific parts of an immune pathway. For example, that hemocytes make enzymes that drive the biochemical melanization cascade whereas the fat body makes proteins that regulates the pathway. But I have a question on how these comparisons were made, and in fact, much more detail is needed regarding how the RNAseq data were analyzed. It seems to me that the RNAseq data is proportional for each tissue, and hence, a direct comparison between cell types is more complicated than presented. For example, a particular mRNA may be enriched in hemocytes and not in fat body, but if there is 1000 times the amount of RNA in fat body than in hemocytes (because fat body cells are bigger and more numerous) then an mRNA that is overabundant in hemocytes may be fewer in total number than the same mRNA when it is underrepresented in fat body. Overall, more detail on the methods of the RNAseq analysis is needed, as is addressing the specific point above.
  3. Section 2.4: In the title of the section, the word to use is “effectively”. But I also caution against using trachea as a cell type, or mentioning that trachea take up nanobeads (unless the nanobeads cross the trachea and are now in the lumen). Tracheae are the respiratory structures, and they are chitinous. Several cell types are attached to the trachea. These include cells that make and maintain the trachea, as well as hemocytes.
  4. Section 2.4: I am curious to know where the nanobeads go by 24 hours after injection. The nanobeads are not described in the methods, but I presume they are not digestible. In that case, they should still be around. Are they amassing elsewhere?
  5. The methods do not provide sufficient detail for another scientist to replicate this study. More detail is required. For example, more detail is needed regarding (1) fat body collection, (2) RNAseq analysis and (3) nanobead treatment. Additional attention should be paid to the language in the manuscript, as the manuscript does not flow as smoothly as one generally expects from this research team. 

Author Response

We appreciate the reviewer comments, which have helped us improve the quality of our manuscript. We have addressed these comments by adding detailed information on methods and editing the text. The corresponding corrections are highlighted in red to make it easy to track changes in the resubmitted files.

Comments and Suggestions for Authors

This manuscript contains an RNAseq dataset that suggests that hemocytes and fat body play complementary roles in the immune response. Moreover, the authors use RNAseq and a chemical method to ablate hemocytes to demonstrate that hemocytes are important in controlling viral infections in the hemocoel but not in the midgut. This is an interesting manuscript that advances our understanding of the antiviral response in mosquitoes, and particularly, the antiviral response in the hemocoel. In that sense, this manuscript presents a significant advance in the field. Below I raise some concerns and make some suggestions. The authors should be able to easily respond to these comments.

Comments 1: More detail is needed on tissue collection, and more specifically, how the fat body was collected. Depending on the method, some of the results and conclusions may need to be modified. The methods section does not detail how the fat body was collected, and a detailed protocol must be provided. Figure 1 implies that fat body cells were squeezed out of the abdomen, which I find unlikely. If the entire abdomen was used, then the samples would also contain abdominal muscle, ventral nerve cord, pericardial cells, heart, and other tissues (including sessile hemocytes). It would even contain circulating hemocytes if not perfused beforehand. If “fat body” is really the abdomen, can the percentage of the tissue that is fat body be calculated? How can the investigators be sure that their findings are attributable to fat body and not other cells. I am fairly confident that fat body is overabundant on the abdominal wall, but more detail is needed.

Response 1: We added the protocol for fat body isolation:  (Lines:502-511). We also added a new reference, where the fat body has been recovered from mosquito hearts.

  1. Cardoso-Jaime, V.; Maya-Maldonado, K.; Tsutsumi, V.; Hernández-Martínez, S. Mosquito Pericardial Cells Upregulate Cecropin Expression after an Immune Challenge. Developmental & Comparative Immunology 2023, 147, 104745, doi:10.1016/j.dci.2023.104745.

Comments 2: I am excited about the potential that hemocytes and fat body each specialize on specific parts of an immune pathway. For example, that hemocytes make enzymes that drive the biochemical melanization cascade whereas the fat body makes proteins that regulates the pathway. But I have a question on how these comparisons were made, and in fact, much more detail is needed regarding how the RNAseq data were analyzed. It seems to me that the RNAseq data is proportional for each tissue, and hence, a direct comparison between cell types is more complicated than presented. For example, a particular mRNA may be enriched in hemocytes and not in fat body, but if there is 1000 times the amount of RNA in fat body than in hemocytes (because fat body cells are bigger and more numerous) then an mRNA that is overabundant in hemocytes may be fewer in total number than the same mRNA when it is underrepresented in fat body. Overall, more detail on the methods of the RNAseq analysis is needed, as is addressing the specific point above.

Response 2:  We have included more detailed information about the RNA-seq analysis (Lines: 516-544). Regarding the purity of hemocyte and fat body cell samples, we cannot estimate this using our current approach. However, we are confident that the samples were highly enriched for either hemocytes or fat body. This is supported by the data shown in Figure 1e, where hemocyte samples display high expression of classical hemocyte markers compared with fat body samples, as expected (Lines 88-91).

Comments 3: Section 2.4: In the title of the section, the word to use is “effectively”. But I also caution against using trachea as a cell type, or mentioning that trachea take up nanobeads (unless the nanobeads cross the trachea and are now in the lumen). Tracheae are the respiratory structures, and they are chitinous. Several cell types are attached to the trachea. These include cells that make and maintain the trachea, as well as hemocytes.

Response 3: We agree that several cell types can associated with the trachea, which can uptake the nano-beads. We have changed the word trachea for tracheal structures or trachea associated cell type. (Lines 243, 247-251, 263)

Comments 4: Section 2.4: I am curious to know where the nanobeads go by 24 hours after injection. The nanobeads are not described in the methods, but I presume they are not digestible. In that case, they should still be around. Are they amassing elsewhere?

Response 4: Most previous phagocytosis studies in mosquitoes have used latex beads of a 1–2 µm diameter, whereas this is the first time nanobeads of this size have been tested in mosquitoes. While it would be interesting to study their long-term fate, our aim here was to mimic the physical properties of DV (Lines: 247-251) and track which tissues could be permissive to viral entry at short time frame.

We apologize for the confusion in the Methods. The nanobeads procedure was originally described in the section titled “Fluorescent assays and microscopy”; however, we have now changed the title to 4.8. Fluorescent bead assays and microscopy” to avoid confusion. In addition, we have added more details to this section. (Lines 544-558)

Comments 5: The methods do not provide sufficient detail for another scientist to replicate this study. More detail is required. For example, more detail is needed regarding (1) fat body collection, (2) RNAseq analysis and (3) nanobead treatment. Additional attention should be paid to the language in the manuscript, as the manuscript does not flow as smoothly as one generally expects from this research team. 

Response 5: We have provided the additional requested details.

Round 2

Reviewer 3 Report

Comments and Suggestions for Authors

Since my previous concerns are answered and revised accordingly. I have no further comments.